# OFFLINE BILINGUAL WORD VECTORS, ORTHOGONAL TRANSFORMATIONS AND THE INVERTED SOFTMAX

**Samuel L. Smith, David H. P. Turban, Steven Hamblin & Nils Y. Hammerla**
babylon health
London, UK
{samuel.smith, steven.hamblin, nils.hammerla}@babylonhealth.com
dt382@cam.ac.uk

## ABSTRACT

Usually bilingual word vectors are trained "online". Mikolov et al. (2013a) showed they can also be found "offline"; whereby two pre-trained embeddings are aligned with a linear transformation, using dictionaries compiled from expert knowledge. In this work, we prove that the linear transformation between two spaces should be orthogonal. This transformation can be obtained using the singular value decomposition. We introduce a novel "inverted softmax" for identifying translation pairs, with which we improve the precision @1 of Mikolov's original mapping from 34% to 43%, when translating a test set composed of both common and rare English words into Italian. Orthogonal transformations are more robust to noise, enabling us to learn the transformation without expert bilingual signal by constructing a "pseudo-dictionary" from the identical character strings which appear in both languages, achieving 40% precision on the same test set. Finally, we extend our method to retrieve the true translations of English sentences from a corpus of 200k Italian sentences with a precision @1 of 68%.

## 1 INTRODUCTION

Monolingual word vectors embed language in a high-dimensional vector space, such that the similarity of two words is defined by their proximity in this space (Mikolov et al., 2013b). They enable us to train sophisticated classifiers to interpret free flowing text (Kim, 2014), but they require independent models to be trained for each language. Crucially, training text obtained in one language cannot improve the performance of classifiers trained in another, unless the text is explicitly translated. Increasing interest is now focused on bilingual vectors, in which words are aligned by their meaning, irrespective of the language of origin. Such vectors may drive improvements in machine translation (Zou et al., 2013), and enable language-agnostic text classifiers (Klementiev et al., 2012). They can also be higher quality than monolingual vectors (Faruqui & Dyer, 2014).

Bilingual vectors are normally trained "online", whereby both languages are learnt together in a shared space (Chandar et al., 2014; Hermann & Blunsom, 2013). Typically these algorithms exploit two sources of monolingual text alongside a smaller bilingual corpus of aligned sentences. This bilingual signal provides a regularisation term, which penalises the embeddings if similar words in the two languages do not lie nearby in the vector space. However Mikolov et al. (2013a) showed that bilingual word vectors can also be obtained "offline". Two sets of word vectors in different languages were first obtained independently, and then a linear matrix $W$ was trained using a dictionary to map word vectors from the "source" language into the "target" language. Remarkably, this simple procedure was able to translate a test set of English words into Spanish with 33% precision.

To develop an intuition for these two approaches, we note that the similarity of two word vectors is defined by their cosine similarity, $\cos(\theta_{ij}) = y_i^T x_j / |y_i| |x_j|$. The vectors have no intrinsic meaning, it is only the angles between vectors which are meaningful. This is closely analogous to asking a cartographer to draw a map of England with no compass. The map will be correct, but she does not know which direction is north, so the angle of rotation will be random. Two maps drawn by two such cartographers will be identical, except that one will be rotated by an unknown angle with respect to the other. There are two ways the cartographers could align their maps. They could draw

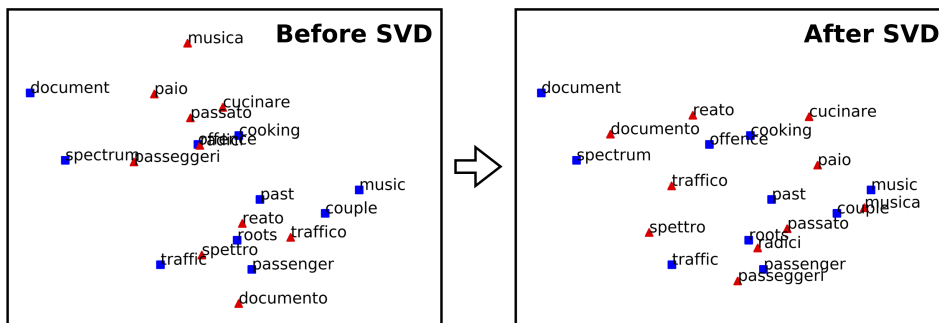

Figure 1: A 2D plane through an English-Italian semantic space, before and after applying the SVD on the word vectors discussed below, using a training dictionary of 5000 translation pairs. The examples above were not used during training, but the SVD aligns the translations remarkably well.

the maps together, thus ensuring that landmarks are placed nearby on both maps during "training". Or they could draw their maps independently, and then compare the two afterwards; rotating one map with respect to the other until the major cities are aligned. We note that the more similar the intrinsic geometry of the two maps, the more accurately this rotation will align the space.

The main contribution of this work is to provide theoretical insights which unify and enhance existing approaches in the literature. We prove that a self-consistent linear transformation between vector spaces should be orthogonal. Intuitively, the transformation is a rotation, and it is found using the singular value decomposition (SVD). The shared semantic space obtained by the SVD is illustrated in figure 1. We build on the work of Dinu et al. (2014), by introducing a novel "inverted softmax" to combat the hubness problem. Using the same word vectors, training dictionary and test set provided by Dinu, we improve the precision of Mikolov's method from 34% to 43%, when translating from English to Italian, and from 25% to 37% when translating from Italian to English. We also present three remarkable new results. First we exploit the superior robustness of orthogonal transformations, by discarding the training dictionary and forming a pseudo-dictionary from the identical character strings which appear in both languages. While Mikolov's method achieves translation precisions of just 1% and 3% respectively with this pseudo-dictionary, our approach achieves precisions of 40% and 34%. This is a striking result, achieved without any expert bilingual signal. Next, we form simple sentence vectors by summing and normalising over word vectors, and we obtain bilingual sentence vectors by applying the SVD to a phrase dictionary formed from a bilingual corpus of aligned text. The transformation obtained aligns the underlying word vectors, achieving a translation precision of 43% and 38%, en-par with the expert word dictionary above. Finally, we show that we can also use our bilingual word vectors to retrieve sentence translations; identifying the translation of an English sentence from a bag of 200k Italian candidate sentences with 68% precision.

## 2 OFFLINE BILINGUAL LANGUAGE VECTORS

### 2.1 PREVIOUS WORK

Offline bilingual word vectors were first proposed by Mikolov et al. (2013a). They obtained a small dictionary of paired words from Google Translate, whose word vectors we denote $\{y_i, x_i\}_{i=1}^{n}$. Next, they applied a linear transformation $W$ to the source language and used stochastic gradient descent to minimise the squared reconstruction error,

$$\min_{W} \sum_{i=1}^{n} ||y_i - W x_i||^2. \tag{1}$$

After training, any word vector in the source language can be mapped to the target by calculating $y_e = W x$. The similarity between a source vector $x$ and a target vector $y_t$ can then be evaluated by the cosine similarity $\cos(\theta_{te}) = y_t^T y_e / |y_t||y_e|$. Astonishingly, this simple procedure achieved 33% accuracy when translating unseen words from English into Spanish, using a training dictionary of

5k common English words and their Spanish translations, and word vectors trained using *word2vec* on the WMT11 text datasets. Translations were found by a simple nearest neighbour procedure.

We note that the cost function above is solved by the method of least squares, as realised by Dinu et al. (2014). They did not modify this cost function, but proposed an adapted method of retrieving translation pairs which was more accurate when translating words from English to Italian. Faruqui & Dyer (2014) obtained bilingual word vectors using CCA. They did not attempt any translation tasks, but showed that the combination of CCA and dimensionality reduction improved the performance of monolingual vectors on standard evaluation tasks. CCA had previously been used to iteratively extract translation pairs directly from monolingual corpora (Haghighi et al., 2008). More recently, Xing et al. (2015) argued that Mikolov's linear matrix should be orthogonal, and introduced an approximate procedure composed of gradient descent updates and repeated applications of the SVD. CCA has been extended to map 59 languages into a single shared space (Ammar et al., 2016), and non-linear "deep CCA" has been introduced (Lu et al., 2015). A theoretical analysis of bilingual word vectors similar to this paper was recently published by Artetxe et al. (2016).

## 2.2 THE SIMILARITY MATRIX AND THE ORTHOGONAL TRANSFORM

To prove that a self-consistent linear mapping between semantic spaces must be orthogonal, we form the similarity matrix, $S = YWX^T$. $X$ and $Y$ are word vector matrices for each language, in which each row contains a single word vector, denoted by lower case $x$ and $y$. The matrix element,

$$\begin{aligned} S_{ij} &= y_i^T W x_j & (2) \\ &= y_i \cdot (W x_j), & (3) \end{aligned}$$

evaluates the similarity between the $j^{th}$ source word and the $i^{th}$ target word. The matrix $W$ maps the source language into the target language. The largest value in a column of the similarity matrix gives the most similar target word to a particular source word, while the largest value in a row gives the most similar source word to a given target word. However we could also form a second similarity matrix $S' = XQY^T$, such that the matrix $Q$ maps the target language back into the source.

$$\begin{aligned} S'_{ji} &= x_j^T Q y_i & (4) \\ &= x_j \cdot (Q y_i), & (5) \end{aligned}$$

also evaluates the similarity between the j$^{th}$ source word and the i$^{th}$ target word. To be self consistent, we require $S' = S^T$. However $S^T = XW^TY^T$, and therefore the matrix $Q = W^T$. If $W$ maps the source language into the target, then $W^T$ maps the target language back into the source.

When we map a source word into the target language, we should be able to map it back into the source language and obtain the original vector. $x \sim W^T y$ and $y \sim W x$ and thus $x \sim W^T W x$. This expression should hold for any word vector $x$ and thus we conclude that the transformation $W$ should be an orthogonal matrix $O$ satisfying $O^T O = I$, where $I$ denotes the identity matrix. Orthogonal transformations preserve vector norms, so if we normalise $X$ and $Y$, then the matrix element $S_{ij} = |y_i||O x_j| \cos(\theta_{ij}) = \cos(\theta_{ij})$. The similarity matrix $S = YOX^T$ computes the cosine similarity between all possible pairs of source and target words under the orthogonal transformation $O$.

We now infer the orthogonal transformation $O$ from a dictionary $\{y_i, x_i\}_{i=1}^n$ of paired words. Since we predict the similarity of two vectors by evaluating $S_{ij} = \cos(\theta_{ij})$, we ought to learn the transformation by maximising the cosine similarity of translation pairs in the dictionary,

$$\max_O \sum_{i=1}^n y_i^T O x_i, \text{ subject to } O^T O = I. \quad (6)$$

The solution proceeds as follows. We form two ordered matrices $X_D$ and $Y_D$ from the dictionary, such that the $i^{th}$ row of $\{X_D, Y_D\}$ corresponds to the source and target language word vectors of the $i^{th}$ pair in the dictionary. We then compute the SVD of $M = Y_D^T X_D = U\Sigma V^T$. This step is highly efficient, since $M$ is a square matrix with the same dimensionality as the word vectors. $U$ and $V$ are composed of columns of orthonormal vectors, while $\Sigma$ is a diagonal matrix containing the singular values. Our cost function is minimised by $O = UV^T$. The optimised similarity matrix,

$$\begin{aligned} S &= YUV^T X^T. & (7) \\ S_{ij} &= y_i^T U V^T x_j & (8) \\ &= (U^T y_i) \cdot (V^T x_j). & (9) \end{aligned}$$

Thus, we map both languages into a single space, by applying the transformation $V^T$ to the source language and $U^T$ to the target language. We prove that this procedure maximises equation 6 in the appendix. It was recently independently proposed by Artetxe et al. (2016), and provides a numerically exact solution to the cost function proposed by Xing et al. (2015), just as the method of least squares provides a numerically exact solution to the cost function of Mikolov et al. (2013a).

Our procedure did not use the singular values $\Sigma$, but these values do carry relevant information. All of the singular values are positive, and each singular value $s_i$ is uniquely associated to a pair of normalised vectors $u_i$ and $v_i$ from the matrices $U$ and $V$. Standard implementations of the SVD return the singular values in descending order. The larger the singular value, the more rapidly the mean cosine similarity of the dictionary decreases if the corresponding vectors are distorted. We can perform dimensionality reduction by neglecting the vectors $\{u_i, v_i\}$ which arise from the smallest singular values. This is trivial to implement by simply dropping the final few rows of $U^T$ and $V^T$, and we will show below that it leads to a small improvement in the translation performance.

## 2.3 THE SVD AND CCA

Our method is very similar to the CCA procedure proposed by Faruqui & Dyer (2014), which can also be obtained using the SVD (Press, 2011). We first obtain the source dictionary matrix $X_D$ and subtract the mean from each column of this matrix to obtain $X'_D$. We then perform our first SVD to obtain $X'_D = Q_D \Sigma_X V_X^T$. We perform the same two operations on the target dictionary $Y_D$ to obtain $Y'_D = W_D \Sigma_Y V_Y^T$, and then perform another SVD on the product $M' = Q_D^T W_D = U' \Sigma' V'^T$. This last step is identical to the alignment procedure we introduced above. Finally we obtain $X'$ and $Y'$ by subtracting the mean value of each column in $X_D$ and $Y_D$, before computing a new pair of aligned representations of the full vocabulary, $Q_{aligned} = X' V_X \Sigma_X^{-1} U'$, and $W_{aligned} = Y' V_Y \Sigma_Y^{-1} V'$. Once again, we perform dimensionality reduction by neglecting the final few columns of $U'$ and $V'$.

Effectively, CCA is composed of two stages. In the first stage, we replace our word vector matrices $X$ and $Y$ by two new vector representations $Q = X' V_X \Sigma_X^{-1}$ and $W = Y' V_Y \Sigma_Y^{-1}$. In the second stage, we apply the orthogonal transformations $\{U', V'\}$ to align $\{Q, W\}$ in a single shared space. To the authors, the first stage appears redundant. If we have already learned high-quality word vectors $\{X, Y\}$, there seems little reason to learn new representations $\{Q, W\}$. Additionally, it is unclear why the transformations $V_X \Sigma_X^{-1}$ and $V_Y \Sigma_Y^{-1}$ are obtained using only the dictionary matrices $\{X_D, Y_D\}$, rather than using the full vocabularies $\{X, Y\}$.

## 2.4 THE INVERTED SOFTMAX

Mikolov et al. (2013a) predicted the translation of a source word $x_j$ by finding the target word $y_i$ closest to $W x_j$. In our formalism, this corresponds to finding the largest entry in the $j^{th}$ column of the similarity matrix. To estimate our confidence in this prediction, we could form the softmax,

$$P_{j \to i} = \frac{e^{\beta S_{ij}}}{\sum_m e^{\beta S_{mj}}}. \tag{10}$$

To learn the "inverse temperature" $\beta$, we maximise the log probability over the training dictionary,

$$\max_\beta \sum_{\text{pairs } ij} \ln(P_{j \to i}). \tag{11}$$

This sum should be performed only over valid translation pairs. Dinu et al. (2014) demonstrated that nearest neighbour retrieval is flawed, since it suffers from the presence of "hubs". Hubs are words which appear as the nearest neighbour target word to many different source words, reducing the translation performance. We propose that the hubness problem is mitigated by inverting the softmax, and normalising the probability over source words rather than target words.

$$P_{j \to i} = \frac{e^{\beta S_{ij}}}{\alpha_j \sum_n e^{\beta S_{in}}}. \tag{12}$$

Intuitively, rather than asking whether the source word translates to the candidate target word, we assess the probability that the candidate target word translates back into the source word. We then select the target word which maximises this probability. If the $i^{th}$ target word is a hub, then the

denominator in equation 12 will be large, preventing this target word from being selected. The vector $\alpha$ ensures normalisation. The sum over $n$ should run over all source words in the vocabulary. However to reduce the computational cost, we only perform this sum over $n_s$ sample words, chosen randomly from the vocabulary. Unless explicitly stated, $n_s = 1500$.

## 2.5 PSEUDO DICTIONARIES

### 2.5.1 IDENTICAL CHARACTER STRINGS

Our method requires a training dictionary of paired vectors, which is used to infer the orthogonal map $O$ and the inverse temperature $\beta$, and also as a validation set during dimensionality reduction. Typically this dictionary is obtained by translating common source words into the target language using Google Translate, which was constructed using expert human knowledge. However most European languages share a large number of words composed of identical character strings. Words like "London", "DNA" and "Tortilla". It is probable that identical strings across two languages share similar meanings. We can extract these strings and form a "pseudo-dictionary", compiled without any expert bilingual knowledge. Below we show that this pseudo dictionary is sufficient to successfully translate between English and Italian with high precision.

### 2.5.2 ALIGNED SENTENCES

The Europarl corpus is composed of aligned sentences in a number of European languages (Koehn, 2005). Chandar et al. (2014) showed that such corpora can be used alongside monolingual text sources to learn online bilingual vectors, but to date, offline bilingual vectors have only been obtained from dictionaries. To learn the orthogonal transformation from aligned sentences, we define the vector $q$ of a source language sentence by a normalised sum over the word vectors present, $q = \sum_i x_i / |\sum_i x_i|$. The vector $w$ of a target language sentence is defined by a normalised sum of word vectors $y_i$. We view the aligned corpus as a dictionary of paired sentences $\{w_i, q_i\}$, from which we form two dictionary matrices $W_D$ and $Q_D$. We obtain the transformation $O$ from an SVD on the matrix $M = W_D^T Q_D$, and use this transformation to translate individual words in the test set.

This simple procedure embeds words and sentences in the same vector space. The sentence embedding can be thought of as the "average word" that the sentence conveys. Intuitively, each aligned sentence pair gives us weak information about a possible word pair in the dictionary. By combining a large number of such sentence pairs, we obtain sufficient information to align the vector spaces and infer the translations of individual words. However, we will go on to show that this orthogonal transformation can be used, not only to retrieve the translations of words between languages, but also to retrieve the translations of sentences between languages with remarkably high accuracy.

## 3 EXPERIMENTS

We perform our experiments using the same word vectors, training dictionary and test dictionary provided by Dinu et al. (2014)[1]. The word vectors were trained using *word2vec*, and then the 200k most common words in both the English and Italian corpora were extracted. The English word vectors were trained on the *WackyPedia/ukWaC* and *BNC* corpora, while the Italian word vectors were trained on the *WackyPedia/itWaC* corpus. The training dictionary comprises 5k common English words and their Italian translations, while the test set is composed of 1500 English words and their Italian translations. This test set is split into five sets of 300. The first 300 words arise from the most common 5k words in the English corpus, the next 300 from the 5k-20k most common words, followed by bins for the 20k-50k, 50k-100k, and 100k-200k most common words. This enables us to evaluate how word frequency affects the translation performance. Some of the Italian words have both male and female forms, and we follow Dinu in considering either form a valid translation.

We report results using our own procedure, as well as the methods proposed by Mikolov, Faruqui, and Dinu. We compute results for Mikolov's method by applying the method of least squares, and results for Faruqui's method using Scikit-learn's implementation of CCA with default parameters. In both cases, we predict translations by nearest neighbour retrieval. We do not apply dimensionality

---

[1]These are available at http://clic.cimec.unitn.it/~georgiana.dinu/down/

Table 1: Translation performance using the expert training dictionary, English into Italian.

| Precision | *Mikolov et al.* | *Dinu et al.* | CCA | SVD | + inverted softmax | + dimensionality reduction |
|---|---|---|---|---|---|---|
| @1 | 0.338 | 0.385 | 0.361 | 0.369 | 0.417 | **0.431** |
| @5 | 0.483 | 0.564 | 0.527 | 0.527 | 0.587 | **0.607** |
| @10 | 0.539 | 0.639 | 0.581 | 0.579 | 0.655 | **0.664** |

Table 2: Translation performance using the expert training dictionary, Italian into English.

| Precision | *Mikolov et al.* | *Dinu et al.* | CCA | SVD | + inverted softmax | + dimensionality reduction |
|---|---|---|---|---|---|---|
| @1 | 0.249 | 0.246 | 0.310 | 0.322 | 0.373 | **0.380** |
| @5 | 0.410 | 0.454 | 0.499 | 0.496 | 0.577 | **0.585** |
| @10 | 0.474 | 0.541 | 0.570 | 0.557 | 0.631 | **0.636** |

reduction following CCA, to enable a fair comparison with our SVD procedure. We compute results for Dinu's method using the source code they provided alongside their manuscript. Their method uses 10k source words as "pivots"; 5k from the test set and 5k chosen at random from the vocabulary. By contrast, the inverted softmax does not know which source words occur in the test set.

## 3.1 EXPERIMENTS USING THE EXPERT TRAINING DICTIONARY

In tables 1 and 2 we present the translation performance of our methods when translating the test set between English and Italian, using the expert training dictionary provided by Dinu. We evaluate Mikolov and Dinu's methods for comparison, as well as CCA- proposed by Faruqui & Dyer (2014). All the methods are more accurate when translating from English to Italian. This is unsurprising given that some English words in the test set can translate to either the male or female form of the Italian word. In the fourth column we evaluate the performance of our SVD procedure with nearest neighbour retrieval. This already provides a marked improvement on Mikolov's mapping, especially when translating from Italian into English. As anticipated, the performance of the SVD is very similar to CCA. In the following two columns we apply first the inverted softmax, and then dimensionality reduction to the aligned vector space obtained by the SVD. The hyper-parameters of these procedures were optimised on the training dictionary. Combining both procedures improves the precision @1 to 43% and 38% when translating from English to Italian, or Italian to English respectively. These results are a significant improvement on previous work. In table 3 we present the dependence of precision @1 on word frequency. We achieve remarkably high precision when translating common words. This performance drops off for rare words; presumably either because there is insufficient monolingual data to learn high quality rare word vectors, or because the linguistic similarities between rare words across languages are less pronounced.

Table 3: Translation precision @1 from English to Italian using the expert training dictionary. We achieve 69% precision on test cases selected from the 5k most common English words in the ukWaC, Wikipedia and BNC corpora. The precision falls for less common words.

| Word ranking by frequency | *Mikolov et al.* | *Dinu et al.* | CCA | SVD | + inverted softmax | + dimensionality reduction |
|---|---|---|---|---|---|---|
| 0-5k | 0.607 | 0.650 | 0.633 | 0.637 | **0.690** | **0.690** |
| 5-20k | 0.463 | 0.540 | 0.477 | 0.510 | 0.580 | **0.610** |
| 20-50k | 0.280 | 0.350 | 0.343 | 0.323 | 0.380 | **0.403** |
| 50-100k | 0.193 | 0.217 | 0.190 | 0.200 | 0.230 | **0.253** |
| 100-200k | 0.147 | 0.163 | 0.163 | 0.173 | **0.203** | 0.200 |

Table 4: Translation performance using the pseudo dictionary of identical character strings.

| Precision | English to Italian: | | | | Italian to English: | | | |
|---|---|---|---|---|---|---|---|---|
| | *Mikolov et al.* | *Dinu et al.* | CCA | This work | *Mikolov et al.* | *Dinu et al.* | CCA | This work |
| @1 | 0.010 | 0.060 | 0.291 | **0.399** | 0.025 | 0.115 | 0.270 | **0.343** |
| @5 | 0.028 | 0.263 | 0.464 | **0.576** | 0.064 | 0.0317 | 0.470 | **0.566** |
| @10 | 0.039 | 0.391 | 0.530 | **0.631** | 0.091 | 0.431 | 0.523 | **0.624** |

## 3.2 EXPERIMENTS USING IDENTICAL CHARACTER STRINGS

In the preceding section, we reported our performance using an orthogonal transformation learned on an expert training dictionary of 5k common English and Italian words. We now report our performance when we do not use this dictionary, and instead construct a pseudo dictionary from the list of words which appear in both the English and Italian vocabularies, composed of exactly the same character string. Remarkably, 47074 such identical character strings appear in both vocabularies. There would be fewer identical entries for more diverse language pairs, but our main goal here is to demonstrate the superior robustness of orthogonal transformations to low quality dictionaries.

We exhibit our results in table 4, where we evaluate our method (SVD + inverted softmax + dimensionality reduction), when translating either from English to Italian or from Italian to English. Even when using this pseudo dictionary prepared with no expert bilingual knowledge, we still achieve a mean translation performance @1 of 40% from English to Italian on our test set. By contrast, Mikolov and Dinu's methods achieve precisions of just 1% and 6% respectively. CCA also performs well, although it became significantly more computationally expensive when the vocabulary size increased. Previously translation pairs have been extracted from monolingual corpora using CCA by bootstrapping a small seed lexicon (Haghighi et al., 2008).

## 3.3 EXPERIMENTS ON THE EUROPARL CORPUS OF ALIGNED SENTENCES

The English-Italian Europarl corpus comprises 2 million English sentences and their Italian translations, taken from the proceedings of the European parliament (Koehn, 2005). As outlined earlier, we can form simple sentence vectors in the word vector space by summing and normalising over the words contained in a sentence. These sentence vectors can be used in two different tasks. First, we can use the Europarl corpus as a training dictionary, whereby the matrices $X_D$ and $Y_D$ are formed from the sentence vectors of translation pairs. By applying the SVD to the first 500k sentences in this "phrase dictionary", we obtain a set of bilingual word vectors from which we can retrieve translations of individual words. We exhibit the translation performance of this approach in table 5. We achieve 42.8% precision @1 when translating from English into Italian and 37.5% precision when translating from Italian into English, comparable to the accuracy achieved using the expert word dictionary on the same test set. It is difficult to compare the two approaches, since they require different training data. However our performance appears competitive with *Bilbowa*, a leading method for learning bilingual vectors online from monolingual corpora and aligned text (Gouws et al., 2015). We do not include results for CCA due to the computational complexity on a dictionary of this size.

Second, we can apply our orthogonal transformation to retrieve the Italian translation of an English sentence, or vice versa. To achieve this, we hold back the final 200k English and Italian sentences from our 500k sample of Europarl, and attempt to retrieve the true translation of a given sentence in this test set. We obtain the orthogonal transformation by performing the SVD on either the expert word dictionary provided by Dinu, or on a phrase dictionary formed from the first 300k

Table 5: Translation performance, using the Europarl corpus as a phrase dictionary.

| Precision | English to Italian: | | | Italian to English: | | |
|---|---|---|---|---|---|---|
| | *Mikolov et al.* | *Dinu et al.* | This work | *Mikolov et al.* | *Dinu et al.* | This work |
| @1 | 0.234 | 0.313 | **0.428** | 0.19 | 0.224 | **0.375** |
| @5 | 0.368 | 0.531 | **0.589** | 0.331 | 0.419 | **0.563** |
| @10 | 0.433 | 0.594 | **0.647** | 0.39 | 0.508 | **0.620** |

Table 6: "Translation" precision @1, when seeking to retrieve the true translation of an English sentence from a bag of 200k Italian sentences, or vice versa, averaged over 5k samples. We first obtain bilingual word vectors, using either the word dictionary provided by Dinu, or by constructing a phrase dictionary from Europarl. We set $n_s = 12800$ in the inverted softmax.

|  | English to Italian: | | Italian to English: | |
|---|---|---|---|---|
|  | Word dictionary | Phrase dictionary | Word dictionary | Phrase dictionary |
| *Mikolov et al.* | 0.105 | 0.166 | 0.120 | 0.206 |
| *Dinu et al.* | 0.453 | 0.406 | **0.489** | 0.459 |
| SVD | 0.268 | 0.431 | 0.473 | **0.656** |
| + inverted softmax | **0.546** | **0.678** | 0.429 | 0.486 |

sentences from Europarl. For simplicity, we do not apply dimensionality reduction here. Our results are provided in table 6. For precision @1, most approaches favour the phrase dictionary, while Dinu's method favours the word dictionary. We show in the appendix that all methods favour the phrase dictionary for precision @5 and @10. Remarkably, given no information except the sentence vectors, we are able to retrieve the correct translation of an English sentence with 67.8% precision. This is particularly surprising, since we are using the simplest possible sentence vectors, which have no information about word order or sentence length. It is likely that we could improve on these results if we used higher quality sentence vectors (Le & Mikolov, 2014; Kiros et al., 2015), although we might lose the ability to simultaneously align the underlying word vector space.

When training the inverted softmax, the inverse temperature $\beta$ diverged, and the "translation" performance from English to Italian significantly exceeded the performance from Italian to English. This suggested that sentence retrieval from Italian to English might be achieved better by nearest neighbours, so we also evaluated the performance of nearest neighbour retrieval on the same orthogonal transformation, as shown in the third row of table 6. This improved the performance from Italian to English from 48.6% to 65.6%, which suggests that the optimal retrieval approach would be able to tune continuously between the conventional softmax and the inverted softmax.

## 4 SUMMARY

We proved that the optimal linear transformation between word vector spaces should be orthogonal, and can be obtained by a single application of the SVD on a dictionary of translation pairs, as proposed independently by Artetxe et al. (2016). We used the SVD to obtain bilingual word vectors, from which we can predict the translations of previously unseen words. We introduced a novel "inverted softmax" which significantly increased the accuracy of our predicted translations. Combining the SVD with the inverted softmax and dimensionality reduction, we improved the translation precision of Mikolov's original linear mapping from 34% to 43%, when translating a test set composed of both common and rare English words into Italian. This was achieved using a training dictionary of 5k English words and their Italian translations. Replacing this training dictionary with a pseudo-dictionary acquired from the identical word strings that appear in both languages, we showed that we still achieved 40% precision, demonstrating that it is possible to obtain bilingual vector spaces without an expert bilingual signal. Mikolov's method achieves just 1% precision here, emphasising the superior robustness of orthogonal transformations. There are currently a number of approaches to obtaining offline bilingual word vectors in the literature. Our work shows they can all be unified.

Finally, we defined simple sentence vectors to obtain offline bilingual word vectors without a dictionary using the Europarl corpus. We achieved 43% precision when translating our test set from English into Italian under this approach, comparable to our results above, and competitive with online approaches which use aligned text as the bilingual signal. We demonstrated that we can also use our sentence vectors to retrieve the true translation of an English sentence from a bag of 200k Italian candidate sentences with 68% precision, a striking result worthy of further investigation.

### ACKNOWLEDGMENTS

We thank Dinu et al. for providing their source code, pre-trained word vectors, and a training and test dictionary of English and Italian words, and Philipp Koehn for compiling the Europarl corpus.

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

## A  THE ORTHOGONAL PROCRUSTES PROBLEM

There is not an intuitive analytic solution to the cost function in equation 6, but an analytic solution does exist to the closely related "orthogonal Procrustes problem", which minimises the squared reconstruction error subject to an orthogonal constraint (Schönemann, 1966),

$$\min_O \sum_{i=1}^{n} ||y_i - Ox_i||^2, \text{ subject to } O^T O = I. \tag{13}$$

However both $X$ and $Y$ are normalised, while $O$ preserves the vector norm. We note that,

$$||y_i - Ox_i||^2 \quad = \quad |y_i|^2 + |x_i|^2 - 2y_i^T Ox_i \tag{14}$$

$$\propto \quad A - y_i^T Ox_i. \tag{15}$$

$A$ is a constant, and so the cost functions given in equations 6 and 13 are equivalent. We presented the solution of the orthogonal Procrustes problem in the main text.

## B  ADDITIONAL EXPERIMENTS

In tables 7 and 8, we provide results at precisions @5 and @10, for the same experiments shown @1 in table 6 of the main text. Once again, the inverted softmax performs well when retrieving the Italian translations of English sentences, but is less effective translating Italian sentences into English. However, the performance of Dinu's method appears to rise more rapidly than other methods as we transition from precision @1 to @5 to @10. Additionally, while Dinu's method performs better when using the word dictionary @1, it prefers the phrase dictionary @5 and @10.

Table 7: "Translation" precision @5, when seeking to retrieve the true translation of an English sentence from a bag of 200k Italian sentences, or vice versa, averaged over 5k samples. We first obtain bilingual word vectors, using either the word dictionary provided by Dinu, or by constructing a phrase dictionary from Europarl. We set $n_s = 12800$ in the inverted softmax.

|  | English to Italian: | | Italian to English: | |
| --- | --- | --- | --- | --- |
|  | Word dictionary | Phrase dictionary | Word dictionary | Phrase dictionary |
| *Mikolov et al.* | 0.187 | 0.272 | 0.221 | 0.326 |
| *Dinu et al.* | 0.724 | 0.732 | **0.713** | 0.765 |
| SVD | 0.394 | 0.546 | 0.619 | **0.774** |
| + inverted softmax | **0.727** | **0.825** | 0.622 | 0.679 |

Table 8: "Translation" precision @10, when seeking to retrieve the true translation of an English sentence from a bag of 200k Italian sentences, or vice versa, averaged over 5k samples. We first obtain bilingual word vectors, using either the word dictionary provided by Dinu, or by constructing a phrase dictionary from Europarl. We set $n_s = 12800$ in the inverted softmax.

|  | English to Italian: | | Italian to English: | |
| --- | --- | --- | --- | --- |
|  | Word dictionary | Phrase dictionary | Word dictionary | Phrase dictionary |
| *Mikolov et al.* | 0.228 | 0.316 | 0.267 | 0.386 |
| *Dinu et al.* | **0.807** | 0.832 | **0.783** | **0.849** |
| SVD | 0.441 | 0.595 | 0.663 | 0.807 |
| + inverted softmax | 0.782 | **0.862** | 0.692 | 0.745 |

