# Peer review of "Offline bilingual word vectors, orthogonal transformations and the inverted softmax"

_ICLR 2017 — accepted_

[Official Review · AnonReviewer1 · rating 6 · confidence 3 · 18 Dec 2016]
originality 4 · substance 3

This paper extends preceding works to create a mapping between the word embedding space of two languages. The word embeddings had been independently trained on monolingual data only, and various forms of bilingual information is used to learn the mapping. This mapping is then used to measure the precision of translations.

In this paper, the authors propose two changes: "CCA" and "inverted softmax".  Looking at Table 1, CCA is only better than Dina et al in 1 out of 6 cases (It/En @1).  Most of the improvements are in fact obtained by the introduction of the inverted softmax normalization.

Overall, I wonder which aspect of this paper is really new. You mention:
 - Faruqui & Dyer 2014 already used CCA and dimensionality reduction
 - Xing et al 2015 argued already that Mikolov's linear matrix should be orthogonal

Could you make clear in what aspect your work is different from Faruqui & Dyer 2014 (other the fact that you applied the method to measure translation precision) ?

Using cognates instead of a bilingual directory is a nice trick. Please explain how you obtained this list of cognates ? Obviously, this only works for languages with the same alphabet (for instance Greek and Russian are excluded)

Also, it seems to me that in linguistics the term "cognate" refers to words which have a common etymological origin - they don't necessarily have the same written form (e.g. night, nuit, noche, Nacht). Maybe, you should use a different term ? Those words are probably proper names in news texts.

[Official Review · AnonReviewer2 · rating 7 · confidence 5 · 19 Dec 2016]
**Inverted Softmax is nice**
originality 3 · impact 3

This paper discusses aligning word vectors across language when those embeddings have been learned independently in monolingual settings. There are reasonable scenarios in which such a strategy could come in helpful, so I feel this paper addresses an interesting problem. The paper is mostly well executed but somewhat lacks in evaluation. It would have been nice if a stronger downstream task had been attempted.

The inverted Softmax idea is very nice.

A few minor issues that ought to be addressed in a published version of this paper:

1) There is no mention of Haghighi et al (2008) "Learning Bilingual Lexicons from Monolingual Corpora.", which strikes me as a key piece of prior work regarding the use of CCA in learning bilingual alignment. This paper and links to the work here ought to be discussed.
2) Likewise, Hermann & Blunsom (2013) "Multilingual distributed representations without word alignment." is probably the correct paper to cite for learning multilingual word embeddings from multilingual aligned data.
3) It would have been nicer if experiments had been performed with more divergent language pairs rather than just European/Romance languages
4) A lot of the argumentation around the orthogonality requirements feels related to the idea of using a Mahalanobis distance / covar matrix to learn such mappings. This might be worth including in the discussion
5) I don't have a better suggestion, but is there an alternative to using the term "translation (performance/etc.)" when discussing word alignment across languages? Translation implies something more complex than this in my mind.
6) The Mikolov citation in the abstract is messed up

[Official Review · AnonReviewer3 · rating 8 · confidence 5 · 19 Dec 2016]
soundness 2 · impact 3 · substance 2

The paper focuses on bilingual word representation learning with the following setting:

1. Bilingual representation is learnt in an offline manner i.e., we already have monolingual representations for the source and target language and we are learning a common mapping for these two representations.
2. There is no direct word to word alignments available between the source and target language.

This is a practically useful setting to consider and authors have done a good job of unifying the existing solutions for this problem by providing theoretical justifications. Even though the authors do not propose a new method for offline bilingual representation learning, the paper is significant for the following contributions:

1. Theory for offline bilingual representation learning.
2. Inverted softmax.
3. Using cognate words for languages that share similar scripts.
4. Showing that this method also works at sentence level (to some extent).

Authors have addressed all my pre-review questions and I am ok with their response. I have few more comments:

1. Header for table 3 which says “word frequency” is misleading. “word frequency” could mean that rare words occur in row-1 while I guess authors meant to say that rare words occur in row-5.
2. I see that authors have removed precision @5 and @10 from table-6. Is it because of the space constraints or the results have different trend? I would like to see these results in the appendix.
3. In table-6 what is the difference between row-3 and row-4? Is the only difference NN vs. inverted softmax? Or there are other differences? Please elaborate.
4. Another suggestion is to try running an additional experiment where one can use both expert dictionary and cognate dictionary. Comparing all 3 methods in this setting should give more valuable insights about the usefulness of cognate dictionary.

[Public Comment · (anonymous) · 22 Dec 2016]
**Theory on "offline" bilingual word vectors and cognates**

Thank you for the interesting paper.
1. Could you elaborate on how your method provides additional theoretical insight into the importance of orthogonality beyond existing work [1] and [2]? We would additionally encourage you to cite [2].
2. In accordance with the review of AnonReviewer1, could you elaborate how your method is different from the existing use of CCA in [3]?
3. As AnonReviewer1 pointed out, cognates are words with the same etymological origin but are usually spelled differently (see [4] for more examples). You should replace this term to make your manuscript more accurate.
4. Given the above points, the main contributions of your paper are a) the inverted softmax and b) the "cognate" dictionary. Is that correct?

[1] Xing, C., Liu, C., Wang, D., & Lin, Y. (2015). Normalized Word Embedding and Orthogonal Transform for Bilingual Word Translation. NAACL-2015, 1005–1010. 
[2] Artetxe, M., Labaka, G., & Agirre, E. (2016). Learning principled bilingual mappings of word embeddings while preserving monolingual invariance. Proceedings of the 2016 Conference on Empirical Methods in Natural Language Processing (EMNLP-16), 2289–2294. 
[3] Faruqui, M., & Dyer, C. (2014). Improving Vector Space Word Representations Using Multilingual Correlation. Proceedings of the 14th Conference of the European Chapter of the Association for Computational Linguistics, 462 – 471.
[4]

[Public Comment · Samuel L Smith · 06 Jan 2017]
**Changes to new version**

Dear reviewers and readers,

We’d like to thank you all for your positive comments about our manuscript. We were particularly pleased that all three reviewers recommended our work be accepted, and by the interest reviewers expressed in the “inverted softmax”. We have uploaded an updated version. There are three main changes we would like to draw readers’ attention to:

Our use of the term “cognates” was misleading and we have removed it from the new version. To be completely clear, we extract the pseudo-dictionary by finding the identical character strings like “DNA” and “Ciao” which appear in both the English and Italian vocabularies. These identical strings can be found trivially without any expert knowledge. 

We also realised that our procedure, while very similar to CCA, is not identical. We apologise for this mistake, which we have corrected in the new version. We believe this realisation strengthens the manuscript. We provide additional experiments, and a discussion of the very close relationship between the methods. The two methods have very similar performance, but our approach is numerically cheaper.

Shortly after our manuscript was submitted to ICLR, another paper was published [1], which presents a similar theoretical analysis of offline bilingual word vectors. We would like to thank the anonymous reader for bringing this work to our attention, now properly cited. This paper also discusses the need for an orthogonal transformation, and proposes the same novel SVD procedure we propose here to obtain this transformation. However, our work contains a number of contributions not present in their work, including:

1.	The use of dimensionality reduction after the SVD
2.	The inverted softmax
3.	The identical strings pseudo-dictionary
4.	Offline vector alignment using a phrase dictionary
5.	Sentence translation retrieval using bilingual vectors

We will respond to the specific comments of each reviewer underneath their reviews.
Best wishes,
Sam

[1] Artetxe, M., Labaka, G., & Agirre, E. (2016). Learning principled bilingual mappings of word embeddings while preserving monolingual invariance. Proceedings of the 2016 Conference on Empirical Methods in Natural Language Processing (EMNLP-16), 2289–2294.

[Public Comment · Samuel L Smith · 13 Feb 2017]
**Final version uploaded**

We have uploaded the final version. The text is unchanged, but we have modified the title to emphasise the aspects of the paper which have been of most interest to readers (particularly the inverted softmax).

We'd like to thank the PC for accepting our manuscript,
Sam

[Final Decision · Program Chairs · 06 Feb 2017]
**ICLR committee final decision**

This is a nice contribution and that present some novel and interesting ideas. At the same time, the empirical evaluation is somewhat thin and could be improved. Nevertheless, the PCs believe this will make a good contribution to the Conference Track.